# TGF-β Isoforms and GDF-15 in the Development and Progression of Atherosclerosis

**DOI:** 10.3390/ijms25042104

**Published:** 2024-02-09

**Authors:** Agnė Liuizė (Abramavičiūtė), Aušra Mongirdienė

**Affiliations:** Department of Biochemistry, Medical Academy, Lithuanian University of Health Sciences, LT-50161 Kaunas, Lithuania; agne.abramaviciute@stud.lsmu.lt

**Keywords:** atherosclerosis, TGF-β, TGF-15, oxLDL, endothelium

## Abstract

The effect of oxidised lipoproteins on the endothelium, monocytes, platelets, and macrophages is a key factor in the initiation and development of atherosclerosis. Antioxidant action, lipoprotein metabolism, and chronic inflammation are the fields of research interest for better understanding the development of the disease. All the fields are related to inflammation and hence to the secretion of cytokines, which are being investigated as potential diagnostic markers for the onset of atherosclerosis. Pathways of vascular damage are crucial for the development of new laboratory readouts. The very early detection of endothelial cell damage associated with the onset of atherosclerosis, allowing the initiation of therapy, remains a major research goal. This article summarises the latest results on the relationship of tumour growth factor beta (TGF-β) isoforms and growth differentiation factor 15 (GDF-15) to the pathogenesis of atherosclerosis: which cells involved in atherosclerosis produce them, which effectors stimulate their synthesis and secretion, how they influence atherosclerosis development, and the relationship between the levels of TGF-β and GDF-15 in the blood and the development and extent of atherosclerosis.

## 1. Introduction

Atherosclerosis is one of the leading causes of cardiovascular disease (CVD) [1]. It is a chronic inflammatory pathological condition affecting the medium and large arteries. Abnormal lipid retention in the intima of the arterial wall leads to the formation of atherosclerotic plaques. Lipid retention stimulates vascular cells to produce inflammatory mediators [2], chemokines [3], and extracellular matrix proteins [4]. In addition, monocytes and platelets produce proinflammatory compounds involved in plaque formation and maintenance of chronic inflammation [3,4]. The effects of oxidised lipoproteins on the endothelium, monocytes, platelets, and macrophages are an important cause of the onset and development of atherosclerosis [2,5]. To better understand the development of atherosclerosis, interest has been focused on antioxidant effects, lipoprotein metabolism, and chronic inflammation. All these areas are related to inflammation and thus to the secretion of cytokines and extracellular matrix proteins, which are being investigated as potential diagnostic markers of atherosclerosis onset. Pathways of vascular injury are crucial for the development of new laboratory readouts.

Our aim is to summarise the state-of-the-art knowledge on the relationship of tumour growth factor beta (TGF-β) isoforms and growth differentiation factor-15 (GDF-15) cytokines to the pathogenesis of atherosclerosis: which cells involved in atherosclerosis produce them, which effectors stimulate their synthesis and secretion, how they affect atherosclerosis development, and the relationship between blood levels of TGF-β and GDF-15 and the development and degree of atherosclerosis. This knowledge will help to summarise their relationship with atherosclerosis and its severity and to guide future research to identify suitable markers for the early detection of atherosclerosis.

## 2. Types of Cytokines

Cytokines are proteins that act in a paracrine manner as communication signals between cells of the immune system and other cells and tissues in the body. They are important for immunity, inflammation, tissue development, repair, and cancer [6]. Cytokines can be produced by every cell in the nucleus of the human body [7] and are divided into several families: interferons (IFN), interleukins (IL), tumour necrosis factors (TNF), colony-stimulating factors (CSF), and transforming growth factors (TGF) [6]. Cytokines can be proinflammatory or anti-inflammatory, depending on their effects [8]. For example, interleukin-1 (IL-1) and tumour necrosis factor-α (TNF-α) exert an acute inflammatory response. Anti-inflammatory cytokines act during recovery from acute inflammation [8]. Not all known cytokines are associated with atherosclerosis and have been studied. Earlier findings on the association of cytokines with atherosclerosis have been reviewed by Dipak P. R. and Thomas S. Davies [9]. In this review, we will focus on the TGF-β superfamily cytokines, TGF-β1,2,3 isoforms and GDF-15, because new information has been discovered in recent years about their role in the development of atherosclerosis.

## 3. TGF-β Family

Transforming growth factor (TGF) is a family of more than 40 proteins, including TGF-β, inhibin, activin, growth differentiation factors (including GDF-15), bone morphogenetic proteins (BMPs), nodal, Mullerian-inhibiting substance, and TGF-β [10]. The human TGF-β superfamily includes 33 genes encoding homodimeric and heterodimeric cytokines [10]. All members of this family share common sequence elements and structural motifs. These proteins promote cell differentiation, division, migration, death, extracellular matrix (ECM) protein production, and adhesion [11,12]. In recent years, the involvement of TGF-β and GDF-15 in atherosclerosis has been intensively analysed.

Angiotensin II (Ang II) has been shown to stimulate TGF-β mRNA expression in VSMCs [13]. It should be mentioned that low doses of Ang II (without increasing blood pressure) have been shown to induce the expression of inflammatory markers and the migration of CD45 cells into the aorta of normotensive mice [14]. In addition, it has been shown that an increase in blood pressure (BP) induces the production of inflammatory cytokines such as interferon-γ, IL-17, and TNF-α in the vascular wall [15]. Thus, this may explain the relationship between the chronic inflammation induced by hypertension (related to increased Ang II concentrations in the blood), leading to endothelial cell (EC) damage and the release of TGF-β related to the development of atherosclerosis. The recently established link between TGF-β1 and vascular stiffness will be discussed in Section 4.

All TGF-β isoforms were found to be expressed as inactive precursors. After proteolytic cleavage, two related parts of TGF-β are synthesised: latent and mature TGF-β. They remain associated with each other and are referred to as the large latent complex (LLC). Latent activity refers to the absence of biological activity [10]. Once released, the LLC bind to the extracellular matrix proteins fibrillin and fibronectin. Three extracellular matrix (fibroblast environment) enzymes are thought to be involved in the active formation of TGF-β: elastase, metalloprotease 2 (MMP2), and tolloid-like family protease (BMP-1) [10]. Endoglin, a type III TGF-β receptor, is involved in the transmission of the TGF-β signal to ECs. TGF-β is known to act through heterotetrameric receptor complexes that signal to cells by (1) activating SMAD proteins that regulate gene transcription [16] and (2) activating non-SMAD pathways that mediate RNA mechanisms [10].

Most in vitro and in vivo studies on the role of TGF-β in atherosclerosis have investigated TGF-β1 or TGF-β without distinguishing between different isoforms. For example, Pei-Yu Chen and co-authors showed that the outcome of TGF-β signalling is cell type dependent. In ECs, TGF-β signalling is a major contributor to atherosclerosis-related vascular inflammation: TGF-β induces the expression of proinflammatory chemokines, cytokines (including C-C motif chemokine ligand-1 (CCL2)), leucocyte adhesion molecules (vascular cell adhesion molecule-1 (VCAM-1) and intercellular adhesion molecule-1 (ICAM-1)), and matrix metalloproteinase -2 (MMP-2) [11]. In vascular smooth muscle cells (VSMCs), TGF-β exerts anti-inflammatory effects. The authors found that inhibition of endothelial TGF-β signalling in hyperlipidaemic mice reduces vascular inflammation, inhibits disease progression, and induces regression of established lesions. The role of TGF-β in cell migration, proliferation, differentiation, extracellular matrix production and adhesion is still controversial [10,12,17,18]. This may be due to the recent identification of different TGF-β subtypes that may not induce the same processes in different cells.

A few years ago, three different isoforms of TGF-β superfamily cytokines were described: TGF-β1, TGF-β2, and TGF-β3. All of them are encoded by unique genes [19]. In recent years, the different roles of the different TGF-β isoforms have started to be investigated. The general knowledge that has been found about the isoforms of the TGF-β family is presented in Table 1.

In the following, we will discuss what has been found in recent years in the field of the influence of individual TGF-β isoforms and GDF-15 on the initiation and development of atherosclerosis.

## 4. Role of TGF-β1 in Atherosclerosis

In most studies, TGF-β1 has been presented as a protective anti-inflammatory cytokine, inhibiting IL-1 β, TNF-α, and interferon-γ, leading to reduced VCAM-1 expression, leukocyte adhesion, and macrophage activity. TGF-β1 is reported to be present in endothelium, vascular smooth muscle cells (VSMCs), macrophages, platelets [20], and haematopoietic cells. It is the best-studied isoform and is considered to be the most important in the cardiovascular system [12]. Some authors have evaluated the relationship between serum TGF-β1 levels and various forms of atherosclerosis.

Ozgur Selim Ser and co-authors studied patients with coronary artery ectasia (CAE). CAE has been identified as a variant of atherosclerosis and is reported to have a similar pathogenesis. The genes involved in TGF-β1 synthesis, secretion, and activation have been the focus of attention. Polymorphisms Rs1800469 (T-509C) and Rs1800470 (Leu10Pro) of the TGF-β1 gene have been implicated in coronary heart disease complications. Rs1800469 (T-509C) is a variation in the promoter region of the TGF-β1 gene that affects the transcriptional activity of the gene and serum TGF-β1 levels. Rs1800470 (Leu10Pro) is a missense polymorphism in the coding sequence of the TGF-β1 gene that results in changes in the amino acid sequence and the secretion of TGF-β1 from the cells [23]. The authors showed that the genotype distribution of the TGF-β1 rs1800469 and rs1800470 polymorphisms does not differ between CAE and healthy individuals. However, TGF-β1 rs1800470 polymorphism levels were lower in patients homozygous for the AA genotype than in the CAE group of patients carrying the G allele (GG+AG genotypes). In addition, the TGF-β1 rs1800469 polymorphism was associated with serum TGF-β1 levels. TGF-β1 was detected at significantly lower levels in CAE patients than in control subjects with normal coronary arteries. Serum TGF-β1 levels < 3980 mcg/dL had a sensitivity of 74% and a specificity of 61% in predicting CAE. The authors suggested that the G allele of TGF-β1 rs1800470 (GG+AG genotypes) and the A allele of TGF-β1 rs1800469 (AA+AG genotypes) may have a protective effect against the development of CAE and concluded that a decrease in blood levels of TGF-β1 may lead to a reduction in the anti-inflammatory effect and an increase in the rate of CAE development [24]. These findings are in line with the results of a previous study, which reported that the rs1800469 and rs1800470 polymorphisms of TGF-β1 are associated with the development of restenosis after coronary stent placement and coronary artery disease [25]. Accordingly, plasma TGF-β1 levels were found to be significantly increased in patients with CAE and coronary artery disease (CAD) compared with patients with CAD alone [26]. This may indicate an association between TGF-β1 and the development of CAE. The level of coronary artery occlusion in the CAE and CAD groups was not reported, which would be useful for assessing the degree of atherosclerosis in the studied groups. The age of all participants is also not provided. The correlation between TGF-β1 and extracellular matrix proteins in the vessel wall could confirm the involvement of TGF-β1 in wall thinning or thickening.

J. Ahmadi and colleagues found that platelet-borne and soluble TGF-β1 levels are higher in CAD patients compared to healthy controls (30 patients and 30 healthy volunteers) [20]. The amount of mature TGF-β1 was eight-fold higher in the platelets of patients. Importantly, soluble (or mature) TGF-β1 was much lower (11–25 kDa), whereas plasma was deficient in latent TGF-β1 (48–63 kDa). Patients’ platelets also had significantly higher levels of latent TGF-β1 than controls. Moreover, platelet-active TGF-β1 levels correlated with P-selectin (r = 0.7299, *p* < 0.0001), CD40L (r = 0.6397, *p* = 0.001), C-reactive protein (CRP, r = 0.5483, *p* = 0.002), and erythrocyte sedimentation rate (ESR, r = 0.5296, *p* = 0.003) in CAD [20]. These correlations may indicate proinflammatory and pro-aggregatory properties and their relationship with platelet active TGF-β1 levels. Significantly higher platelet expression of active TGF-β1 was also found in CAD patients compared with healthy subjects. In addition, CD40L has been shown to induce oxidative stress, endothelial dysfunction and upregulation of metalloproteases, tissue factors, and chemokines [27]. All of these factors are involved in the progression of atherosclerosis, and TGF-β1 may be involved in this process.

Despite the view of some researchers that TGF-β1 is a protective cytokine in the early stages of atherosclerosis (AS), the opposite has been previously suggested for its effects in the late stages [28]. Zhao and co-authors recently investigated the association between blood TGF-β1 levels and ultrasound-determined AS grade (Table 2). They found a negative correlation of TGF-β1 levels with AS grade. The authors’ findings suggest that TGF-β1 may serve as a quantitative indicator of AS severity [28]. In the present study, current AS was diagnosed by measuring carotid intima-media thickness (CIMT). It should be borne in mind that the study groups were highly inhomogeneous. Patients with carotid artery AS were divided into three groups to assess the association of blood TGF-β1 levels with the degree of AS: (1) the increased intima-media thickness (IMT) group (n = 27), (2) the plaque group (n = 40), and (3) the control group (n = 30). All groups included individuals with hypertension and diabetes mellitus and individuals without hypertension and diabetes mellitus. The scores of the plaques were determined as the sum of the maximum IMT of each plaque. As both hypertension [29,30] and diabetes mellitus [16] play a role in TGF-β1 levels in the blood, they may have biased the results. To determine the association of blood TGF-β1 levels with the degree of AS, more precisely selected cohorts should be studied.

Some studies have examined the relationship between TGF-β1 and blood pressure and vascular stiffness. E. Nakao and co-authors found that elevated plasma TGF-β1 levels may predict the development of hypertension in normotensive subjects [29]. They found a positive correlation between TGF-β1 and left ventricular mass index. It is not possible to say whether elevated levels of TGF-β1 are the result or the cause of cardiac hypertrophy. Other studies have shown that TGF-β1 can stimulate renin activity [30], correlated with systolic blood pressure and vascular age [31]. These correlations increase the importance of TGF-β1 in blood pressure regulation.

One study investigated the effect of TGF-β1 on endothelium and monocytes in type 2 diabetes mellitus (T2DM). It showed that an imbalance in monocyte function has consequences for vascular healing and can lead to atherosclerosis. To assess the impact of TGF-β1 on endothelial resistance to growth factors and monocyte dysfunction, L.M. Makowski and co-authors examined CD14++CD16− monocytes from 41 patients without T2DM and 24 individuals with T2DM. The authors found that T2DM enhances the migratory response of monocytes to a low concentration TGF-β1 gradient. The results suggest that TGF-β1 may influence monocyte function and contribute to vascular complications in T2DM patients [16]. However, it seems that the results should be verified in a more carefully selected group of subjects, as smoking, dyslipidaemia, and obesity were present in both groups and may have influenced the results. Accordingly, the influence of TGF-β1 on other monocyte subsets (CD14++/CD16+, CD14+/CD16++) involved in inflammatory reactions and phagocytosis was not revealed [32].ijms-25-02104-t002_Table 2Table 2Role of TGF-β1 in atherosclerosis.ReferenceInvestigated MediumPatients’ GroupsAge (Year)Stenosis DegreeNumber of CasesResultJ. Ahmadi et al., 2023[20]PRPCAD/healthy59.9 ± 7.575–95%30/30There are 2 different TGF-β1 bands: 48–63 kDa for latent and 11–20 kDa for active. The soluble TGF-β1 levels were significantly higher in patients. Platelet active TGF-β1 correlated with proinflammatory/pro-aggregatory markers.Ozgur Selim Ser et al., 2021 [24]TGF-β1 rs1800469 and rs1800470 variations were investigated in isolated total DNA from EDTA blood sampleCAE/person with normal coronary arteries and typical symptoms of ischaemiaNot presentedNot presented56/44Serum TGF-β1 levels were significantly lower in CAE patients. TGF-β1 rs1800470 G allele carriers had higher TGF-β1 concentrations than patients with the AA genotype in the CAE group.V. I. Podzolkov et al., 2021 [31]Blood plasmaCAH/unCAH/healthy40–70 yearsStenosis of brachiocephalic arteries < 50%80/30/30TGF-β1 levels were significantly higher in the unCAH group and higher in the CAH group compared to the healthy group. TGF-β1 was correlated with CAVI (r = 0.733, *p* = 0.0001) and SBP (r = 0.469, *p* = 0.0001) in all groups.H. Zhao et al., 2017[28]Blood serumHealthy/carotid AS/age-matched controls without carotid AS30–74 yearsPatients with ultrasound-diagnosed carotid artery AS but normal serum lipid profile, FBG, and ALT levels60/67/30TGF-β1 levels negatively correlated with age (r = −0.318, *p* = 0.013) and Crouse scores (r = −0.393, *p* < 0.05). E. Nakao et al., 2017[29]Blood plasmaNormotensive person64.1 ± 9.4No stenosis149 persons followed up 14 yearsSBP (*p* = 0.002) and smoking (*p* = 0.026) significantly associated with TGF-β1 levels in the blood. L.-M. Makowski et al., 2021[16]CD14++CD16- monocytes Healthy/T2DM patients58.61 ± 9.04No stenosis41/24High glucose enhances TGF-β signalling in primary monocytes by inducing TGF-β1 ligand and T βRII receptor gene expressionK. Terada et al., 2018[33]ApoE−/− mice (B6.129P2-Apoetm1Unc N11)Periaortic adipose tissue surrounding the descending thoracic aorta was removed from 16-week-old male mice (C57BL/6NCrSlc) and grafted onto endogenous infrarenal abdominal periaortic adipose tissue from 20-week-old apoE−/− mice that had been fed a high-cholesterol diet for 12 weeks16-week-old and 20-week-old male mice
Not presentedTGF- β 1 levels were elevated in the serum of tPAT-transfected apoE−/− mice. The levels of anti-inflammatory cytokines such as TGF-b1, IL-4, and IL-10 were significantly increased in the transplanted tPAT compared to the donor tPAT, accompanied by an increase in the expression of the proinflammatory cytokines TNF-α and IL-6. TGF-β 1 was expressed by alternatively activated macrophages.PRP—platelet-rich plasma, CAD—stable coronary artery disease, CAE—coronary artery ectasia, CAH—controlled arterial hypertension, unCAH—uncontrolled arterial hypertension, CAVI—cardio-ankle vascular index, SBP—systolic blood pressure, T2DM—type 2 diabetes mellitus, tPAT—periaortic fat pad surrounding the descending thoracic aorta, AS—atherosclerosis, FBG—fasting blood glucose.

The results of the studies relating TGF-β1 levels in blood or supernatant to the degree or development of atherosclerosis are summarised in Table 2. Recent work has shed some light on TGF-β1: (1) There are two bands of TGF-β1 in the human body: latent and active. Only the active form is secreted into the blood. (2) There are different data on the levels of TGF-β1 in the blood of CAD patients: some studies have found elevated levels, and some have found the opposite. (3) The correlation between TGF-β1 and blood pressure, age, and AS severity has been found in heterogeneous groups of patients. (4) High glucose levels enhance TGF-β1 signalling in primary monocytes. It seems that TGF-β1 levels in the blood may depend on the degree of atherosclerosis and should be investigated in further work. All valid human studies should be considered pilot studies due to the small number of cases, and the results should be replicated in larger and more carefully selected patient groups. The groups of patients must be free of comorbidities. The relationship between TGF-β1 and blood pressure should be investigated to see if the increase in TGF-β1 in hypertensive patients is a cause or a consequence.

## 5. Role of TGF-β2 and TGF-β3 in Atherosclerosis

There are few recent studies on the association of TGF-β2 with atherosclerosis. One of them analyses the influence of TGF-β2 on plaque stability and another on the influence of TGF-β2 on the process of endothelial–mesenchymal transition (EndMT).

High levels of TGF-β2 in plaque were associated with plaque stability and a lower risk of future acute cardiovascular (CV) events. TGF-β2 has been shown to be a key component of asymptomatic plaques and to correlate with vascular smooth muscle cell (VSMC) content [19]. Moreover, low levels of TGF-β2 in plaque are associated with age-predicted CV events, suggesting a protective role against atherosclerotic complications, which has not been previously investigated. TGF-β2 was found to be a key component in the differences between asymptomatic and symptomatic plaque. The finding of reduced levels of TGF-β2 in plaques is consistent with the previous finding that TGF-β2 determines plaque stability. TGF-β2 has been found to be associated with VSMC count and increased collagen content in plaque [19]. Collagen has been shown to provide plaque stabilisation [34], and enzyme matrix metalloproteinase 9 (MMP-9) has been shown to catalyse the degradation of collagen and is associated with plaque rupture [35]. A. Edsfeldt et al. showed that TGF-β2 inhibits MMP-9 activity, measured in the cell culture supernatants. Reduced activity of MMP-9 should lead to reduced collagen degradation in the plaque. Accordingly, TGF-β2 in combination with TGF-β3 was confirmed to inhibit inflammation. The biological effect of TGF-β2 should be greater because its concentration was found to be 20-fold higher than TGF-β3 in the plaque [19].

It is important that CCL2 gene expression and CCL2 secretion from THP-1 are pre-stimulated by TGF-β2 before activation of the cell line with lipopolysaccharides (LPSs). This is consistent with the finding that TGF-β2 levels in the plaque correlated negatively with CD68 (a monocyte/macrophage marker) expression, confirming that the reduced CCL2 levels resulted in reduced migration of myeloid cells into the plaque. An important finding was that patients with TGF-β2 levels in the highest quartile had a lower risk of CV events. Another finding suggests that TGF-β2 is predominantly secreted by VSMCs in human atherosclerotic lesions [19]. What drives VSMCs to secrete TGF-β2 is still unknown. Although TGF-β2 was assayed in the plaque, it was not measured in the blood. Moreover, the plaques used in the study were obtained from patients with advanced atherosclerosis and may not be representative of the early stages of the disease. Some of the correlations obtained are not strong and caution may be needed in their interpretation. Asymptomatic patients tend to have a higher degree of stenosis (>80%) compared to symptomatic patients (>70%), and risk factors differ between the groups: hypertension and smoking were more common in the symptomatic group. In conclusion, more rigorous studies in different groups with different degrees of atherosclerosis are needed in the future to clarify the association of TGF-β2 with plaque formation.

TGF-β2 was previously found to be the only TGF-β isoform with a gradual increase during the endothelial-to-mesenchymal transition (EndMT). EndMT is a biological process in which ECs acquire a fibroblastic phenotype, with a concomitant loss of apical–basal polarity and intracellular junction proteins [22]. The TGF-β signalling pathway is presented as an important molecular pathway responsible for the activation of EndMT. Their data suggest that a microenvironment at the interface between inflammation (IL-1β) and tissue remodelling (TGF-β2) can strongly stimulate EndMT. IL-1β and TGF-β2 synergistically promote EndMT. TGF-β2 is a more potent inducer of EndMT than TGF-β1 [21]. All isoforms of TGF-β have been found to be expressed in atherosclerosis, but TGF-β2 has been shown to be more effective in inducing EndMT [21]. There is a lack of data investigating the different functions of TGF-β isoforms in the induction of EndMT. During the development of atherosclerosis, EndMT can be induced by several TGF-β isoforms. In recent years, there has been limited research on EndMT.

Experiments in which human microvascular endothelial cells (HMECs) were treated with different isoforms of TGF-β showed that only TGF-β2 significantly increased the levels of mesenchymal transcription factors and, in combination with IL-1β, promoted EndMT [36].

One study investigated the molecular mechanisms and signalling pathways involved in EndMT induced by TGF-β2 in macrovascular ECs obtained from different types of endothelial cells (ECs) [22] (Table 3). Four types of endothelial cells (primary aortic endothelial cells (PAECs); coronary artery endothelial cells (CAECs); human pulmonary artery endothelial cells (HPAECs); and human umbilical vein endothelial cells (HUVECs)) were stimulated with 10 ng/mL of TGF-β2. PAECs showed the best response to the TGF-β2 treatment, displaying phenotypic changes such as loss of endothelial marker and acquisition of mesenchymal markers, which are consistent with the EndMT activation. The authors concluded that the anatomical origin of ECs influences their ability to undergo EndMT.

When endothelial cells are exposed to impaired flow, the expression of semaphorin 7A (Sema7A) is upregulated or increased. The semaphorin family consists of membrane-bound proteins that are originally located on immunocytes. Sema7A has been implicated in T cell-mediated inflammation and has been shown to promote atherosclerosis [37] by mediating endothelial dysfunction and monocyte–EC interactions [38]. Sema7A is found in atherosclerotic lesions and is associated with EndMT [39]. One group of investigators found that endothelial expression of Sema7A induces EndMT and, in turn, promotes TGF-β2 transcription. The secreted TGF-β2 activates the TGF-β2/Smad signalling pathway and subsequently induces EndMT [40].

The work on TGF-β2 is summarised in Table 3. In brief, recent work has shown that (1) TGF-β2 may be more effective in inducing EndMT, and induction may depend on anatomical origin, (2) TGF-β2 may be predominantly secreted from VSMCs in atherosclerosis, and (3) a high level of TGF-β2 in the plaque may be associated with plaque stability and a lower risk of future CV events. Future work is needed to determine the anatomical origin of EndMT formation, to assess the ability of not only VSMCs but also ECs, macrophages, and monocytes to secrete TGF-β2 in the cell line, and to test the association of blood TGF-β2 levels with different degrees of atherosclerosis in both animal and human models.ijms-25-02104-t003_Table 3Table 3Role of TGF-β2 in atherosclerosis.ReferenceInvestigated MediumInvestigated GroupsAge Number of CasesResultLei Hong et al., 2020[40]Human umbilical vein endothelial cells (Sema7A-HUVECs)Sema7A−/− mice.Not presentedNot presentedTGF-β2 activates TGF-β2/Smad signalling pathway and subsequently induces EndMT.F. U. Ferreira et al., 2019 [22]Different types of endothelial cells (CAECs; PAECs; HUVECs; and HPAECs)


The anatomical origin of ECs influences their ability to undergo EndMT.A. Edsfeldt et al., 2023[19]Human carotid plaques, THP-1 macrophagesPatients undergoing endarterectomy.In vitro experiments: THP-1 macrophages. Symptomatic carotid plaque + stenosis >70% or without symptoms +>80% stenosis 70% or without symptoms +>80%Not presented223 plaquesTGF-β2 was the most abundant TGF-β form in human carotid plaques and the main component in asymptomatic plaques. It is associated with increased levels of collagen and vascular smooth muscle cells in plaque. High levels of TGF-β2 in the plaque associate with reduced risk of future CV events. Negatively associated with MMP-9. TGF-β2 associated with reduced plaque inflammation activity in vitro. J. Zhang et al., 2021[36]Human aortic endothelial cells (HAECs), aortic sinus of miceMice after 12 weeks with a Western dietNot presentedNot presentedTGF-β2 promotes EndMT. TGF-β2 attenuates cell invasion in EndMT in vitro model. CV—cardiovascular, EndMT—endothelial-to-mesenchymal transition, HUVECs—human umbilical vein endothelial cells, CAECs—coronary artery endothelial cells, PAECs—primary aortic endothelial cells, HUVECs—human umbilical vein endothelial cells, HPAECs—human pulmonary artery endothelial cells.

TGF-β3 is thought to affect wound healing by reducing fibrosis and scarring at the wound site [41]. The reduction of fibrosis seems to be beneficial for the regression of atherosclerotic plaque. However, since TGF-β subtypes started to be studied separately, only two papers we found analysed TGF-β3 (Table 4). One of these is the work of A. Edsfeldt and co-authors discussed above. They found that TGF-β3 levels in plaques correlated with MMP-2 (0.311, *p* = 0.00003) and were not associated with a lower risk of CV events [20]. Another study investigating the role of T helper cells in the development of atherosclerotic plaques in a murine model showed that TGF-β3 levels were reduced in supernatants derived from T cells and plasma from atherogenic mice [42]. TGF-β3 is thought to be an anti-inflammatory cytokine. It seems that measurement of TGF-β3 blood levels could be informative for the assessment of atherosclerosis. The cells that secrete TGF-β3, the factors that stimulate its secretion, and the levels of TGF-β3 in the blood of atherosclerotic patients and healthy people have not yet been investigated.

## 6. Growth Differentiation Factor 15

The TGF-β family includes growth differentiation factor 15 (GDF-15), also named macrophage inhibitory cytokine-1. General knowledge about GDF-15 synthesis, receptors, and mechanisms of acting are well discussed by J. Wischhusen and co-authors [43]. In short, GDF-15 synthesis in the cells is induced under stress conditions. Predecessors of GDF-15 involve signal peptide, prodomain, and mature GDF-15 parts. Intracellular proteolytic processing of the predecessor is possible, but GDF-15 is mostly released from the cells as a proprotein. The prodomain remains attached to the extracellular matrix. Activation of GDF-15 is mediated by different proprotein convertases depending on the tissue. The molecular weight of active GDF-15 is low—25 kD per dimer—and it has a half-life of about 3 h in humans due to renal clearance. GDF-15 was shown to signal through glial cell-derived neurotrophic factor (GDNF) family α-like (GFRAL) receptors in cancer cells and brainstem neurons, but clearly, GDF-15 signalling in cells related to atherosclerosis development has not been explored yet.

In humans, GDF-15 is expressed by cardiomyocytes, ECs, macrophages, VSMCs, and adipose tissue [44]. Although its biological functions are still under investigation, GDF-15 has been shown to be involved in the regulation of vascular remodelling and development in atherosclerosis: it regulates the differentiation, proliferation, and survival of ECs, macrophages, VSMCs, and T cells [45] and protects against ageing-mediated systemic inflammatory reactions [46]. Its concentration in the blood has been shown to increase with ageing [47]. GDF-15 is independently associated with markers of EC activation (P-, E-selectin, ICAM-1, and VCAM-1). These molecules are involved in the development of atherosclerosis [48]. All of these cells secrete GDF-15 in response to metabolic and oxidative stress and/or stimulation by proinflammatory cytokines [44,49]. IL-1β, TNF-α, and IL-2 are cytokines that activate macrophages, and in response, macrophages release GDF-15 [49]. Despite the view that GDF-15 has both beneficial and detrimental functions, is highly context-dependent, and may vary considerably depending on the stage of the disease, the data summarised by J. Wang and co-authors suggest that GDF-15 is involved in the initiation and progression of atherosclerosis [50] and has been implicated in subclinical atherosclerosis [51]. Previous studies have shown that elevated levels of GDF-15 are associated with the development of atherosclerotic plaques [52]. These data suggest that GDF-15 may be a marker of vascular pathology associated with atherosclerosis.

The studies on the association of GDF-15 with atherosclerosis that have been conducted over the last five years cover several areas: (1) the influence of GDF-15 on macrophage activity, (2) the association of GDF-15 with lipid metabolism, (3) the influence of GDF-15 on VSMCs, and (4) the study of serum levels of GDF-15 in cardiovascular disease (CVD) patients. The results are summarised in Table 5 and Table 6.

OxLDL-treated macrophages increase GDF-15 expression and foam cell formation and affect autophagic processes [52]. Autophagy is important for many physiological and pathological processes, but little is known about the regulation of autophagy in the pathogenesis of atherosclerosis [52]. A. Heduschke and co-authors investigated the effects of GDF-15 on autophagy using an in vitro human atherosclerotic macrophage model and an in vivo GDF-15-deficient (GDF-15−/−) apolipoprotein E knockout (apoE−/−) mouse model of experimental atherosclerosis [53]. In vitro and in vivo experiments have shown that GDF-15 in combination with oxLDL directly increases macrophage activity [52,53]. This suggests the importance of GDF-15 in the development of atherosclerotic plaque. V. Mushenkova and co-authors reviewed that various cytokines, oxidised lipids, and hypoxia in the plaque influence the change in macrophage phenotype to M1 (proinflammatory) or M2 (anti-inflammatory) [54]. M1 macrophages phagocytose oxLDL, whereas M2 macrophages remove cellular debris and apoptotic cells, so it should be useful to reveal which macrophage subtype is involved in autophagic processes. It shows GDF-15 importance in the development of atherosclerotic plaque. 

Another group of studies showed that GDF-15 deficiency reduces 6.7% of pulmonary trunk lumen stenoses in hypercholesterolaemic GDF-15−/−/ApoE−/− mice compared to ApoE−/− mice. It was found that plaque necrosis area was significantly reduced in GDF-15-deficient mice [55]. Furthermore, when atherosclerotic plaques in the pulmonary trunk of GDF-15-deficient ApoE−/− mice were examined after 5 months on a cholesterol-rich diet, a reduced necrotic core area was observed compared with ApoE−/− mice, and the percentage of α-actin+ VSMCs was increased, while the percentage of CD68+ macrophages was higher [55]. These data indicate the deleterious effects of GDF-15 on plaque changes in atherosclerotic mice.

H. Huang and co-authors invested in humans (healthy and suffering from atherosclerosis), mice with an atherosclerosis model, and mouse macrophages (Table 4). They found that GDF-15 levels were significantly higher in patients. In atherosclerotic mice, GDF-15 was highly expressed in plaque nuclei and macrophages. Macrophages may be the main cells producing GDF-15. GDF-15 inhibited lipid accumulation and reduced inflammation in oxLDL-treated macrophages. In addition, GDF-15 was able to inhibit the oxLDL-induced inflammatory response (including the downregulation of IL-6, IL-8, MCP-1, and MMP-9 in oxLDL-treated macrophages), which promotes the initiation and development of atherosclerosis. These data suggest that GDF-15 may inhibit the initiation and progression of atherosclerosis. The results suggest that GDF-15 has a variety of protective effects against atherosclerosis, including inhibition of monocyte infiltration into the arterial wall and inhibition of plaque development and growth [45]. These results suggest that GDF-15 may inhibit the accumulation of lipoproteins in macrophages and the activation of inflammation, leading to the inhibition of plaque formation and atherosclerosis. Furthermore, it could be a target that could be used to study the development of atherosclerosis.

Another group of studies found that oxLDL in rat VSMCs induces GDF-15 expression, as well as in macrophages treated with oxLDL [56]. OxLDL has previously been shown to induce cell proliferation in a concentration- and time-dependent manner [57]. Furthermore, GDF-15 expression was found to be increased in oxLDL-activated macrophages and significantly correlated with MMP-9. These correlations may reflect the effect of macrophage activation on plaque development [58].

A summary of recent findings using animal and cellular models is presented in Figure 1. In summary, GDF-15 was able to increase autophagic activity and oxLDL-independent lipid accumulation and inhibit lipid accumulation in oxLDL-treated macrophages. OxLDL was able to decrease GDF-15 expression in macrophages and induce GDF-15 expression in VSMCs. GDF-15 could induce VSMC proliferation in vitro. Future studies are needed to confirm that GDF-15 promotes VSMC proliferation in vivo. The effect of GDF-15 on inflammation and lipid accumulation should be confirmed in future experiments using a GDF-15 inhibitor. Further work is needed to unravel the relationship between GDF-15 expression in macrophages, the type of macrophages capable of expressing GDF-15, the amount of GDF-15 in the supernatant in animal and cellular models, and the different levels of atherosclerosis to further evaluate the role of GDF-15 in the pathogenesis of atherosclerosis.ijms-25-02104-t005_Table 5Table 5Relationship of GDF-15 to atherosclerosis in animal and cell models.ReferenceInvestigated Medium/MethodologyInvestigatedGroupsAge Number of CasesResultA. Heduschke et al., 2021 [53]Human monocyte line THP-1, GDF-15-deficient (GDF-15−/−) mice, apoE−/− mice model. THP-1 cells were differentiated to MΦ by PMA and were treated with human rGDF-15 and siGDF-15.GDF-15−/−/ApoE−/− andApoE−/− mice were fed for 20 weeks with a CEDAt the age of 10 weeks12/12siGDF-15 reduced and rGDF-15 increased the autophagic activity in MF. GDF-15−/−/ApoE−/− mice, after CED showed reduced lumen stenosis in the BT.K. Ackermann et al., 2019 [52]Human monocyte line THP-1 was differentiated to MF by PMA and treated with oxLDL + rGDF-15 or human rGDF-15 for 4 h. 
Not presentedNot presentedoxLDL reduces GDF-15 expression in MF.It seems that rGDF-15 promotes autophagosome formation under oxLDL conditions. GDF-15 enhances oxLDL-independent lipid accumulation in human MF.GDF-15 silencing in human MF inhibits oxLDL-induced lipid accumulation. GDF-15/oxLDL impairs autophagy with consequences for lipid homeostasis in human MF.H. Huang et al., 2020 [45](1) GDF-15, IL-6, IL-8, MCP-1, and MMP-9 were measured in serum using ELISA kits. (2) C57BL/6 mice and ApoE−/− mice with a C57BL/6 background were fed a high-fat diet for 5 months. rGDF-15 (50 mg/kg/d) was intravenously injected into ApoE−/− mice after 10 weeks of high-fat feed once every three days. Aorta with plaques was collected after 5 months.(3) Mice MΦ was treated with oxLDL or oxLDL + GDF-15 for 2 days, and supernatant was collected. Healthy/patients with atherosclerosis6-week-old mice65/101 personGDF-15 levels were significantly higher in patients. GDF-15 suppressed lipid accumulation in oxLDL-treated MF. GDF-15 contributed to a decreased inflammatory response in oxLDL-treated macrophages.Y Sun et al., 2019 [56]GDF-15 level in VSMC cell culture media was measured.VSMC from Wistar ratsNot presented
GDF-15 was upregulated in VSMCs after the treatment with oxLDL in vitro. MF—macrophage, siGDF-15—transiently silenced GDF-15, rGDF-15—recombinant GDF-15, CED—calories cholesterol-enriched diet, BT—brachial trunk, PMA—phorbol 12-mystriate 13-acetate, ECs—endothelial cells.
ijms-25-02104-t006_Table 6Table 6Relationship of GDF-15 to human atherosclerosis.ReferenceInvestigated Medium/MethodologyInvestigatedGroupsAge Number of CasesResultH. Kaiser et al., 2021 [51]PlasmaPatients with a history of moderate-to-severe psoriasis with/without atherosclerotic CVD.59.1 (11.0) years39/46GDF-15 levels were increased in patients with CVD compared to those without CVD. GDF-15 is negatively associated with vascular inflammation in the ascending aorta and positively associated with CIMT and CCS in patients without CVD. GDF-15 is associated with the presence of plaques in both carotid arteries.Y. Shimizu et al., 2023 [59]SerumGeneral population of normal-weight older Japanese with normal thyroid function.60–69 years old536A positive association between serum GDF-15 and atherosclerosis was found in both smoking and nonsmoking groups. The same relationship was found in both male and female groups. A sex-adjusted correlation was found between age and GDF-15 levels in participants aged ≥ 65 years.Wei Wang et al., 2020 [60]PlasmaPatients with intermediate CAD (20–70% stenosis in more than 1 main coronary branch by coronary angiography). Patients’ follow-up 7 years: with CV event and event-free.18–80 years old134/407GDF-15 significantly correlated with age, smoking, and SBP. No associations with CV risk factors. Patients with CV events have higher GDF-15 levels than those in event-free group. There was an increase in the MACE during the follow-up for the patients with high levels of GDF-15. S. Hassanzadeb et al., 2021 [61]SerumCAD and control groups. CAD was confirmed by coronary angiography: in CAD group, at least 1 main coronary vessel > 50% luminal narrowing, in controls—no or <50% luminal narrowing.20–80 years old88/88GDF-15 levels were significantly higher in patients with CAD compared to patients without CAD.J.B. Euchouffo-Tcheugui et al., 2021 [62]Blood serum3792 from 4 U.S. communities.80 (SD:5) years old
GDF-15 was positively associated with atherosclerotic CAD and markers of myocardial stress or injury. A.Efat et al., 2022 [63]Blood serum Patients with beta-thalassemia and healthy controls. Subclinical atherosclerosis was evaluated by CIMT.27.17 ± 5.7560/30GDF-15 levels were higher in patients than in controls. GDF-15 was positively correlated with CIMT, total cholesterol, age, and hs-CRP.L. Royston et al., 2022 [64]Blood serumPatients with CVD living with HIV for at least 15 years and without HIV.Over the age of 40 years95/52Increased GDF-15 levels are strongly associated with the presence of coronary artery plaques independently of CV risk factors.CVD—cardiovascular disease, CIMT—carotid intima-media thickness, CCS—coronary artery calcium score, CAD—coronary artery disease, SBP—systolic blood pressure, CV—cardiovascular, BMI—body mass index, MACE—major adverse cardiac event, hs-CRP—high-sensitivity C-reactive protein, HIV—human immunodeficiency virus.

Some work has investigated the levels of GDF-15 in the blood of different groups of people in relation to atherosclerosis. H. Kaiser and co-authors studied 85 patients with moderate to severe psoriasis with or without atherosclerotic CVD [51]. They showed that GDF-15 levels were elevated in patients with CVD compared with patients without CVD. Among patients with psoriasis and without CVD, GDF-15 levels were negatively associated with vascular inflammation and positively associated with carotid artery intima-media thickness (CIMT) and coronary calcium score (CCS). In addition, GDF-15 was positively associated with the American College of Cardiology/American Heart Association (ACC/AHA) risk score and with CCS, after adjusting for the ACC/AHA risk score and hs-CRP. Thus, GDF-15 was associated with all subclinical CVD outcomes examined, including both negative (vascular inflammation) and positive (CIMT and CCS) associations. The possible protective role of GDF-15 in plaque formation may be consistent with the negative association between subclinical vascular inflammation and GDF-15 levels found in this study. However, it cannot be excluded that elevated GDF-15 levels in patients with CVD may be a consequence but not a cause of CVD. Moreover, the different results may depend on the less accurate imaging techniques used to determine the phenotypes of inflammation and atherosclerosis. The limitations of this study (small sample size, absence of a control group without psoriasis, different psoriasis activity of the patients, and the fact that half of them were taking systemic treatment for psoriasis) do not allow us to validate the use of GDF-15 as a marker of atherosclerosis in subclinical psoriasis patients.

Results from a 7-year follow-up of patients with moderate CVD showed that blood levels of GDF-15 were associated with subsequent cardiovascular events, which persisted after controlling for known risk factors and may be related to subclinical atherosclerosis [60]. The data confirmed that GDF-15 is a marker of prognosis for CV events in patients with moderate coronary artery lesions. In another study, the optimal cut-off value for GDF-15 was evaluated in 176 subjects with and without CAD and was found to be 1233 ng/L, with a specificity and sensitivity for CAD of 71% [61]. However, to better understand the link between blood levels of GDF-15 and cardiovascular disease, a genetic predisposition to higher levels of GDF-15 needs to be assessed.

As GDF-15 has been considered a potential marker of [47], another group of researchers investigated the association between GDF-15 and CIMT-confirmed atherosclerosis in a normal-weight elderly Japanese population [59]. They found a significant positive association between GDF-15 and atherosclerosis. The significance did not change after adjusting for thyroid function and known CVD risk factors. The associations were the same in men and women. The positive correlation between current smokers and GDF-15 levels became slightly stronger among nonsmokers. Sex-adjusted significant correlations between age and GDF-15 levels were found only in participants aged ≥ 65 years. Recent work has shown a correlation of GDF-15 with age in different age and condition groups: those with moderate CVD aged 18–80 years [60] and those with beta-thalassaemia aged 21–32 years [63]. Thus, it remains unclear whether GDF-15 in healthy individuals correlates only with age ≥ 65 years and whether the correlations with beta-thalassaemia are influenced by patient condition.

Clinical studies show that elevated levels of GDF-15 lead to a poorer prognosis in patients with CVD. Higher CIMT is associated with angiogenesis [65]. Since mitochondrial stress increases GDF-15 levels [66], GDF-15 promotes VSMC proliferation [56], oxLDL has been shown to be associated with GDF-15 action [45,52,53,55], and GDF-15 levels correlate with CIMT [51], GDF-15 may be associated with the degree of atherosclerosis. As GDF-15 expression on ECs has been shown to be induced by C-reactive protein (CRP) [67] and atherosclerosis is chronic inflammation, future studies should assess CRP levels to confirm the association between GDF-15 and the degree of atherosclerosis.

Previous studies have shown that GDF-15 is increased in hypertension and atherosclerosis [49,50,58,68]. In addition, GDF-15 has been found to correlate with age in healthy [59] and unhealthy individuals [60,63]. Recent studies have also shown a correlation between GDF-15 and CRP, suggesting a link between inflammation and GDF-15 [60,61]. The results reflect the complex relationship of GDF-15 with myocardial, adipose tissue and vascular metabolism [62]. Indeed, GDF-15 is associated with vascular injury, inflammation, and apoptosis [45,51,61]. It should be mentioned that some studies have found a negative association between GDF-15 and vascular inflammation in the ascending and entire aorta in patients with psoriasis [51]. The positive association of GDF-15 with atherosclerosis-induced CVD [59,60,61] and subclinical atherosclerosis [51], and the findings that GDF-15 may act as an EC activator promoting both lesion-associated and normal angiogenesis [69] support the notion that GDF-15 could be considered as one of the potential contributors to atherosclerosis development.

ECs, macrophages, and VSMCs secreted GDF-15 in response to metabolic and oxidative stress and stimulation by the cytokines IL-1β, TNF-α, and IL-2. These cytokines activate macrophages and macrophages secrete GDF-15 in response. GDF-15 can induce VSMC proliferation, inhibit lipid accumulation, and reduce oxLDL-induced inflammation by decreasing the levels of IL-6, IL-8, MCP-1, and MMP-9 in macrophages. The reduction in MCP-1 inhibits monocyte infiltration into the vascular wall and inhibits plaque growth (not shown). OxLDL may decrease GDF-15 expression in macrophages and induce GDF-15 expression in VSMCs.

In summary, serum GDF-15 concentrations in healthy subjects of all ages and in patients with varying degrees of atherosclerosis are not well defined. The mechanism underlying the positive association between serum GDF-15 levels and atherosclerosis at different ages is not yet clear. The influence of environmental and risk factors on GDF-15 levels over the lifetime should be assessed in the future. The mechanisms by which GDF-15 may contribute to the risk of CVD are still not clearly understood.

## 7. Conclusions

Recently, the importance of TGF-β subtypes in the atherosclerotic process, which have started to be studied separately, is starting to reveal contradictions in previous studies on TGF-β, where some authors found it to be related to protecting, while others found it to have a harmful effect on atherosclerosis development. It appears that each TGF-β subtype may act differently depending on the cells affected and the degree of atherosclerosis development. The molecular mechanism of the members of the TGF-β superfamily in plaque formation is welcome. It appears that GDF-15 may be involved in the onset and progression of atherosclerosis and could relate to subclinical atherosclerosis. The very early detection of EC lesions associated with the onset of atherosclerosis, allowing the initiation of therapy, remains an important research field. The usefulness of blood TGF-β isoforms and GDF-15 measurements in the assessment of EC damage and atherosclerosis onset should be further clarified in more carefully selected patient groups.

## Figures and Tables

**Figure 1 ijms-25-02104-f001:**
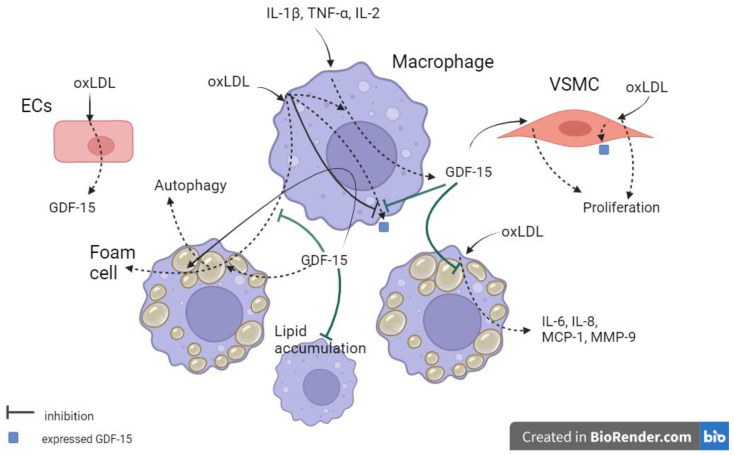
Summarised knowledge of GDF-15 secretion, expression, and action in animal and cell models. ECs—endothelial cells, VSMC—vascular smooth muscle cell, oxLDL—oxidised low-density lipoprotein, dotted lines indicate the result of inducers’ action on the cell, solid lines indicates exposed cells.

**Table 1 ijms-25-02104-t001:** Summarised differences of TGF-β family isoforms.

TGF-β Family Isoforms	TGF-β1	TGF-β2	TGF-β3
Literature	[16,19,20,21]	[17,19,21,22]	[17,19,22]
Abundance of human carotid artery plaques	++	+++, associated with asymptomatic plaque phenotype	+
Relationship to plaque remodelling	No	MMP-2 (r = 0.208) MMP-3 (r = 0.168) MMP-9 (r = −0.310) TIMP-1 (r =−0.227) TIMP-2 (r = 0.309)	MMP-2 (r = 0.311)
Correlation with α-smooth muscle actin in plaque	No	r = 0.262	No
Correlation with calcium area in plaque	r = −0.187	r = −0.154	r = −0.190
Correlation with CD68	No	r = −0.133	No
Secretory cells	Endothelium, VSMC, macrophages, platelets, and hematopoietic cells	VSMC	Not founded
Affected cells have been examined	ECs, monocytes, macrophages	Macrophages, ECs	Not explored
Inducing EndMT	+	+++	Not known
Effect on inflammation/atherosclerotic plaque	Increases or decreases/plaque progression	Suppressed/reduced migration of myeloid cells into plaque	Fibrosis suppression/reduction

MMP—matrix metalloproteinases, TIMP—tissue inhibitors of matrix metalloproteinase, VSMC—vascular smooth muscle cells, EndMT—endothelial to mesenchymal transition, abundance of human carotid artery plaques: + low, ++ intermediate, +++ high.

**Table 4 ijms-25-02104-t004:** The works investigating TGF-β3 in relation to atherosclerosis.

Reference	Investigated Medium	Patients’ Groups	Age	Stenosis Degree	Number of Cases	Result
A.Edsfeldt et al., 2023 [19]	Human carotid plaque	Patients undergoing endarterectomy. In vitro experiments: THP-1 macrophages	Not presented	Symptomatic carotid plaque + stenosis >70% or without symptoms +>80% stenosis 70% or without symptoms +>80%	223 plaques	TGF-β3 level in the plaques was found to correlate with MMP-2 (0.311, *p* = 0.00003) and not associated with a lower risk of CV events.
A.K. Wara et al., 2022 [42]	Aortic plaque. Supernatants from differently treated primary bone marrow-derived macrophages and native T cells isolated from the spleens	CD4-specific KLF10 knockout mice on high-cholesterol diet for 3 months	8-week-old male mice	Not investigated	Not presented	Supernatants derived from T cells and plasma derived from atherogenic mice have increased levels of IFN-γ and reduced levels of TGF-β3.

VSMC—vascular smooth muscle cells, CV—cardiovascular, MMP—matrix metalloproteinase.

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
