# Peer review of "TGF-β Isoforms and GDF-15 in the Development and Progression of Atherosclerosis"

_ijms, 2024, doi:10.3390/ijms25042104_

Round 1

Reviewer 1 Report

Comments and Suggestions for Authors

all my comments are included in the text

Author Response

Thanks to the Reviewer for carefully reviewing the article and helpful comments. We have corrected the article in the light of the comments. It improved the quality of the article. The following corrections have been made. Please see comments in the attached document.

Reviewer 2 Report

Comments and Suggestions for Authors

Comments:

 1. Abstract: English require attention, line# 16-17, “This article summarises the state-of-the-art on on the relationship of”?

2. Section 3. TGF-β family: line#70-71,Angiotensin II (Ang II) has been shown to increase c gene expression’, which c-gene (c-fos, c-jun??), reference 13 not supporting the statement, no data available on c gene expression in the cited reference. Review should be written coherently, it looks fragmented. Text not described adequately or not linked appropriately. Line#78, ‘chapter 4’ can be replaced by ‘section 4’.

3. Table 1 title ‘TGF-β family isoforms’ can be revised, make it more informative as per the content of the table. Take care of English ‘migration myeloid cells’.

4.  Line#117, Use standard abbreviations with proper symbols “IL-1b, TNF-a”, Comment applies to the entire manuscript.

5. Line#133-134, serum TGF-β1 rs1800470 polymorphism levels were lower’ very misleading, revised appropriately.

6. Line#241, ‘leading to reduced collagen degradation in the plaque in vitro in a THP-1 macrophage assay’, how collagen degradation in the plaque in vitro established? Explain the findings with clarity/Reframe sentence appropriately.

7. Title is not in agreement with content. Contents should address those studies with clinical correlation with diseases development/progression and severity. Title is more about the involvement of TGF-β and GDF-15 in the pathogenesis of atherosclerosis. Title can be modified. There are many recent literature associated with biomarkers that may be incorporated and discussed for e.g. Rostan et al.,2022 https://doi.org/10.3389/fcvm.2022.964650

8. GDF-15 is superficially explained, the mechanism of action should be thoroughly explained. Manuscript not even explained its receptor? glial cell-derived neurotrophic factor receptor-alpha-like (GFRAL)? Is there any role of GFRAL in atherosclerosis.

9. Conclusion: line 507-511, looks complex and confusing. Revise appropriately and simplified manner.

Comments on the Quality of English Language

English require attention and should be improved as per the above mentioned comments.

Author Response

Thanks to the Reviewer for carefully reviewing the article and helpful comments. We have corrected the article in the light of the comments. It improved the quality of the article. Please see the following corrections have been made in attached document.
